# The genomic landscape of tuberous sclerosis complex

Katie R. Martin[1], Wanding Zhou[2], Megan J. Bowman[3], Juliann Shih[4], Kit Sing Au[5], Kristin E. Dittenhafer-Reed[1], Kellie A. Sisson[1], Julie Koeman[6], Daniel J. Weisenberger[7], Sandra L. Cottingham[8], Steven T. DeRoos[9], Orrin Devinsky[10], Mary E. Winn[3], Andrew D. Cherniack[4], Hui Shen[2], Hope Northrup[5], Darcy A. Krueger[11] & Jeffrey P. MacKeigan[1,12]

Tuberous sclerosis complex (TSC) is a rare genetic disease causing multisystem growth of benign tumours and other hamartomatous lesions, which leads to diverse and debilitating clinical symptoms. Patients are born with *TSC1* or *TSC2* mutations, and somatic inactivation of wild-type alleles drives MTOR activation; however, second hits to *TSC1/TSC2* are not always observed. Here, we present the genomic landscape of TSC hamartomas. We determine that TSC lesions contain a low somatic mutational burden relative to carcinomas, a subset feature large-scale chromosomal aberrations, and highly conserved molecular signatures for each type exist. Analysis of the molecular signatures coupled with computational approaches reveals unique aspects of cellular heterogeneity and cell origin. Using immune data sets, we identify significant neuroinflammation in TSC-associated brain tumours. Taken together, this molecular catalogue of TSC serves as a resource into the origin of these hamartomas and provides a framework that unifies genomic and transcriptomic dimensions for complex tumours.

[1] Center for Cancer and Cell Biology, Van Andel Research Institute, 333 Bostwick Avenue NE, Grand Rapids, Michigan 49503, USA. [2] Center for Epigenetics, Van Andel Research Institute, 333 Bostwick Avenue NE, Grand Rapids, Michigan 49503, USA. [3] Bioinformatics and Biostatistics Core, Van Andel Research Institute, 333 Bostwick Avenue NE, Grand Rapids, Michigan 49503, USA. [4] Cancer Program, Broad Institute of Harvard and MIT, 415 Main Street, Cambridge, Massachusetts 02142, USA. [5] Department of Pediatrics, University of Texas Health Science Center at Houston-McGovern Medical School, 6431 Fannin, Houston, Texas 77030, USA. [6] Cytogenetics and Pathology Core, Van Andel Research Institute, 333 Bostwick Avenue NE, Grand Rapids, Michigan 49503, USA. [7] Norris Comprehensive Cancer Center, University of Southern California, 1450 Biggy Street, Los Angeles, California 90033, USA. [8] Department of Pathology, Spectrum Health System, 100 Michigan Street NE, Grand Rapids, Michigan 49503, USA. [9] Division of Pediatric Neurology, Helen DeVos Children's Hospital, Spectrum Health System, 100 Michigan Street NE, Grand Rapids, Michigan 49503, USA. [10] Department of Neurology, New York University School of Medicine, 223 E 34 Street, New York, New York 10016, USA. [11] Division of Neurology, Cincinnati Children's Hospital Medical Center, 3333 Burnet Avenue, Cincinnati, Ohio 45229, USA. [12] College of Human Medicine, Michigan State University, 220 Trowbridge Road, East Lansing, Michigan 48824, USA. Correspondence and requests for materials should be addressed to J.P.M. (email: mackeig1@msu.edu).

Tuberous sclerosis complex (TSC) is a neurocutaneous, autosomal dominant genetic disease affecting ∼1 in 6,000 to 10,000 live births[1–4]. TSC causes highly variable, multisystem growth of benign tumours and other hamartomatous lesions that cause diverse clinical problems[5]. Abnormal brain growths are one of the most common features of TSC and lead to epilepsy, developmental delay, cognitive impairment, autism, behavioural problems and hydrocephalus. Most prevalent of these are cortical tubers, largely static malformations of the cerebral cortex that are present at birth and associated with seizure activity[3,6]. Approximately 80% of patients develop subependymal nodules (SENs) on the lateral ventricle walls, which can progress into subependymal giant cell astrocytomas (SEGAs), larger well-circumscribed tumours near the foramen of Monro. Several studies have suggested an association between brain lesions and neurological symptoms in TSC patients, underscoring the need to understand and reduce these growths to improve quality of life[7–10].

Other major organs affected by TSC lesions are the skin, kidney, lung and heart. Skin lesions include hypomelanotic macules and facial angiofibromas that are important diagnostic features of TSC and affect nearly all TSC patients. Renal angiomyolipomas (RAs) affect more than 70% of patients and are typically benign lesions that can cause kidney dysfunction and require treatment if significantly large, abundant or susceptible to bleeding[11]. In fact, RAs are the most common cause of mortality in adult TSC patients[12]. Finally, heart tumours called cardiac rhabdomyomas (CRMs) are another major diagnostic feature of TSC, as they can be detected prenatally and are most common in infants.

While TSC may be inherited (familial), it is more often the result of de novo (sporadic) germline mutations in one of two tumour suppressor genes, TSC1 (encoding TSC1, also known as hamartin) and TSC2 (encoding TSC2 or tuberin)[13–15]. Purely heterozygous germline mutations as well as mosaic mutations have been identified in TSC patients[16–19]. Along with TBC1D7, TSC1 and TSC2 form a physical complex that supports the GTPase-activating protein (GAP) activity of TSC2 towards the small GTPase, RHEB, a direct and positive regulator of MTOR (specifically, MTOR complex I or MTORC1)[20]. MTORC1 integrates signals from growth factors, amino acids and energy to promote cell growth, division and survival. Accordingly, loss-of-function mutations in TSC1 or TSC2 lead to constitutive MTORC1 activation that is uncoupled from upstream signalling inputs. This molecular insight led to the evaluation of MTOR inhibitors in clinical trials and U.S. Food and Drug Administration (FDA) approval for therapeutic use in TSC patients[21–25]. Despite considerable promise, MTOR inhibitors are not universally effective across the TSC population, fail to maintain tumour reduction following cessation of treatment, and may be associated with undesirable side effects[23,25,26]. Therefore, a critical need remains to develop additional therapeutic options for TSC, including those that target tumour growth.

While TSC lesions may develop by somatic inactivation of TSC1/TSC2, second hits (mutations) are not always observed, especially in brain lesions, suggesting that additional mechanisms may contribute to their growth[27–32]. Moreover, the collective molecular changes underlying TSC tumour growth are unknown, yet essential to understanding disease aetiology and developing therapies. To address this, we implement a comprehensive genomics study to characterize the molecular landscape of TSC. We evaluate 111 TSC-associated tissues for TSC1/TSC2 status, DNA mutations, copy number aberrations, differential gene expression and DNA methylation patterns. We find that unlike a majority of RAs and SEN/SEGAs, only one-third of cortical tubers are driven by somatic TSC1/TSC2 inactivation, suggesting

monoallelic mutation may be sufficient to cause cortical malformation. Further, we discover that most TSC lesions have a low somatic mutational burden, in contrast to malignant tumours. Instead, large arm-level chromosomal aberrations are found in tumours from a subset of patients (11%). We uncover conserved gene expression signatures for each lesion type, and use computational cell sorting to identify individual components of pleiotropic tumours. Moreover, we identify a substantial immune expression signature in TSC-associated brain tumours, particularly SEN/SEGAs, which is supported by immunohistochemistry. Taken together, this study provides a comprehensive genomic landscape of TSC, knowledge around cell-of-origin, and unifies the molecular signatures of these complex tumours.

## Results

**TSC1 and TSC2 mutational spectrum.** Genomic DNA and total RNA were isolated from 78 fresh-frozen TSC lesions, including 31 cortical tubers (TUB), 20 RAs, 20 SEN/SEGA (2 SEN and 18 SEGA, which represent a continuum of the same tumour), 5 CRMs and 2 skin lesions. In addition, 33 TSC-associated non-tumour tissues and 16 non-TSC (normal) brain and kidney tissues were included. As genetic material permitted, samples were assayed on the following platforms: whole-exome sequencing (WES), Illumina Infinium Omni2.5 single-nucleotide polymorphism (SNP) arrays, Illumina Infinium HumanMethylation450 (HM450) BeadArrays, targeted-deep TSC1/TSC2 sequencing and mRNA sequencing (RNAseq). Sample name prefixes correspond to patient identifiers and suffixes indicate individual tissue samples (for example, 01-RA1 denotes RA sample 1 from patient 01). All available sample information is presented in Supplementary Data 1.

Our first objective was to characterize the mutational spectrum of TSC1 and TSC2 (refs 2,14). For this, we used WES to define point mutations (single-nucleotide variants or SNVs) and small insertions or deletions (INDELs), and high-resolution SNP arrays to identify large deletions and regions of copy-neutral loss-of-heterozygosity (CN-LOH). We also suspected that a small fraction of TSC1/TSC2 mutations may be missed by these two platforms including systemic mosaic or subclonal somatic mutations occurring at low allelic frequencies, medium-sized deletions beyond the capture of WES and absent from greater scale copy number segmentation based on SNP arrays, and mutations found in regions of poor coverage (for example, splicing mutations within introns). To address this, we implemented targeted-deep sequencing of the entire TSC1 and TSC2 loci—including upstream and downstream elements, introns and exons—to better detect such mutations. Collectively, we identified 57 unique DNA mutations (SNVs and INDELs) in TSC2 and eight in TSC1 from both normal and lesion tissues (Fig. 1a and Supplementary Data 2). Most mutations were observed in only a single individual with the exception of two TSC2 mutations, which were shared by two or more unrelated patients. Mutations were distributed across each locus with no enrichment in specific domains or hotspots[33]. Targeted sequencing identified point mutations in non-tumour tissue from three patients that were not detected by WES. These included a heterozygous germline splicing mutation at an exon–intron boundary in TSC1 (62-UG1), and two apparently mosaic mutations found at <5% allelic frequency (in patients 57 and 74). We also identified two low-frequency somatic TSC2 mutations in tumour tissues: a frameshift mutation in 18-RA1 and nonsense mutation in 06-RA1. The latter of these co-occurred with a somatic TSC2 mutation already identified by WES, suggesting independent second hits drove subclonal growth

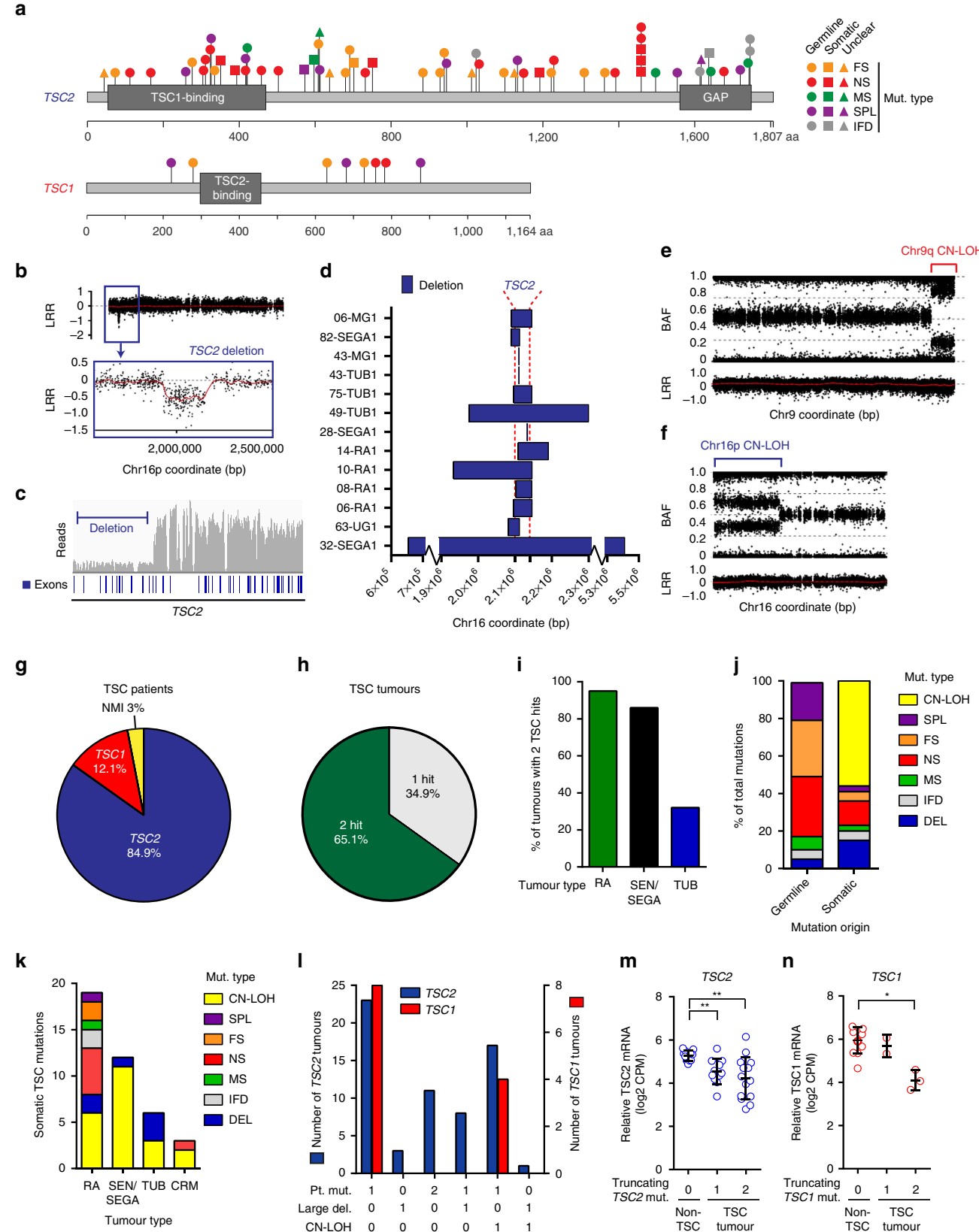

within the tumour. This approach also allowed fine-mapping of deletions first observed by SNP array, and enabled detection of intragenic deletions below the limits of detection by SNP array including a deletion spanning a single exon in patient 43

(Supplementary Fig. 2). Using SNP arrays, we found larger (>100 bp) deletions in *TSC2* in eleven patients, ranging in size from 462 bp to 4.8 Mb, with a median size of 48.6 kb (Fig. 1b–d and Supplementary Data 2). Furthermore, we used SNP arrays to

detect regions of CN-LOH, identified as areas in which B-allele frequencies (BAF) diverge from the heterozygous state while copy numbers (log-R ratios) remain stable (Fig. 1e,f and Supplementary Data 2). These affected *TSC2* (chromosome 16p13) in 18 lesions, and *TSC1* (chromosome 9q34) in 4. Last, we used HM450 arrays to assess genome-wide DNA methylation profiles with a focus on the promoters and gene bodies of *TSC1/TSC2*. We did not find evidence of epigenetic silencing of *TSC1* or *TSC2* in any tissue (Supplementary Fig. 1).

Taken together, we identified *TSC1/TSC2* mutations in 64 of 66 (97%) patients (84.9% *TSC2* and 12.1% *TSC1*), leaving two patients (3%) with no mutation identified (NMI), a smaller percentage than previous estimates based on conventional molecular testing[15,33] (Fig. 1g). We found no mutations in other MTOR pathway genes in these two NMI patients, nor did we find any genes with both germline and somatic variants, supporting the hypothesis that a third TSC-causative locus ('*TSC3*') does not exist.

As TSC lesions are thought to arise by Knudson's two-hit model of tumorigenesis, we next specifically investigated the germline and somatic origin of the *TSC1/TSC2* mutations occurring in these tissues. For lesions lacking patient-matched normal samples, we made predictions for the somatic or germline origin of mutations (indicated by asterisks in Supplementary Data 2; see Methods) based primarily on allele frequencies. We discovered that roughly two-thirds of hamartomas from *TSC1/TSC2* patients harboured two TSC hits, including most RAs and SEN/SEGAs, while second hits were found in only 35% of cortical tubers (Fig. 1h,i). Frameshift INDELs and splicing mutations rarely occurred somatically, despite representing over half of germline mutations. Instead, CN-LOH events, which arise from errors in mitotic recombination, were the most common type of second hit and nearly the exclusive somatic event in SEN/SEGAs (Fig. 1j,k). For both *TSC1* and *TSC2*, lesions with single point mutations or point mutations in combination with CN-LOH were most common, although *TSC2* lesions with two point mutations and combinations involving large deletions were also found, in contrast to *TSC1* (Fig. 1l). Finally, we wanted to determine whether TSC1 and TSC2 expression was decreased in lesions with mutations predicted to decrease or truncate transcripts. Despite considerable heterogeneity, tumours with one or two truncating mutations in *TSC2* showed reduced levels of TSC2 mRNA transcripts compared to non-TSC tissues (pair-wise Welch's *t*-tests; FDR-adjusted $P = 0.01$) (Fig. 1m). Similarly, tumours with two truncating *TSC1* mutations showed a lower level of TSC1 mRNA compared to non-TSC tissue (pair-wise Welch's *t*-tests; FDR-adjusted $P = 0.03$) (Fig. 1n).

**Coding mutational landscape of TSC tumours is quiet.** In addition to *TSC1/TSC2*, we hypothesized that lesions may acquire mutations in other genes, including those that affect tumour growth. To test this, we profiled the coding genome of 42 lesions paired with normal samples using WES. We uncovered a median somatic mutation rate of 0.31 mutations per megabase (Mb) of DNA (range: 0.16–3.8 mutations per Mb), including silent and non-silent SNVs and small INDELs, with a median variant allelic fraction (VAF) of 0.13 (Fig. 2a and Supplementary Data 3). This mutation rate is substantially lower than almost all malignant tumour types, with the exception of acute myeloid leukaemia (AML) (Fig. 2b). We found that 10 of 42 (24%) tumours contained at least one somatic mutation in a candidate or high-confidence tumour driver gene[34], although there was no enrichment in tumours lacking somatic TSC1/TSC2 inactivation (that is, tumours with less than two *TSC1/TSC2* mutations) (Supplementary Data 3). Moreover, no specific mutations recurred across patients and only two genes were somatically mutated in more than one patient. Importantly, we also failed to find somatic mutations in any other MTOR pathway gene.

**Subset of TSC tumours harbour large chromosomal aberrations.** Given this low mutational burden, our next objective was to determine whether large chromosomal copy number aberrations (CNAs) exist that may play a role in tumour development. Aside from deletions and CN-LOH events involving *TSC1/TSC2*, we discovered that nine lesions from eight TSC patients harboured large (arm or whole chromosome level) CNAs at other chromosomal locations (Fig. 2c,d and Supplementary Datas 4 and 5). This included chromosome 1 and chromosome 12 CNAs in four tumours each, and chromosomes 5, 7, 11, 17 and 19 CNAs in two tumours each. The remaining CNAs were not shared across multiple tumours. These CNAs were found in each of the major lesion types studied (RA, TUB, SEN/SEGA), as well as CRM, and not found in any normal (non-lesion) tissues. Five of these CNA-bearing tumours also showed *TSC1/TSC2* CN-LOH, and in all cases, a larger fraction of DNA was affected by the CN-LOH event than these CNAs, suggesting they occurred subsequently to a driving LOH event. Importantly, we used fluorescent *in situ* hybridization (FISH) on fresh-frozen tumour sections to confirm 24 of 25 (96%) molecularly-detected arm-level events (Fig. 2e and Supplementary Table 1).

**RAs display adipose and PEComa features.** Our next goal was to define the molecular signatures of each TSC hamartomatous lesion type using genome-wide DNA methylation and transcript profiling. Unsupervised clustering of DNA methylation array data revealed lesions of each type clustered with one another and away from normal (non-TSC) tissue counterparts, suggesting a high degree of molecular conservation within each (Supplementary Fig. 3). To investigate tumour-specific methylation, we calculated the hypermethylation fraction of each lesion as the fraction of probes methylated that lack methylation in a panel of normal tissues. Among TSC lesions, RAs had the highest hypermethylation fraction (Fig. 3a). Although this level was just a fraction of the hypermethylation observed in malignant tumours

**Figure 1 | The mutational spectrum of *TSC1* and *TSC2* in TSC patient samples.** (**a**) SNVs and INDELs in *TSC2* and *TSC1* (each mutation shown only once per patient). (**b**) Large *TSC2* deletion in 10-RA1 identified by SNP array. Black dots: log-R ratios (LRR) of probe intensities; lower panel: magnification of boxed region. (**c**) Intragenic *TSC2* deletion in 63-UG1 identified by targeted sequencing; grey bars: read counts along *TSC2*; blue bars: exons. (**d**) Blue bars: large deletions in *TSC2*. X-axis is broken to accommodate 32-SEGA1 deletion. (**e,f**) Example CN-LOH events as involving *TSC1* on chromosome 9q in 38-SEGA1 (**e**) and *TSC2* on chromosome 16p in 27-SEGA1 (**f**). (**g**) Percentage of 66 TSC patients with a mutation in *TSC1*, *TSC2* or neither (NMI). (**h**) Percentage of tumours with 1 or 2 *TSC1/TSC2* mutations identified. (**i**) Percentage of *TSC1/TSC2* mutant tumours with 2 hits. Only 1-hit lesions with all 3 DNA platforms completed included (Supplementary Data 2). (**j**) Percentage of germline and somatic *TSC1/TSC2* mutations of each type. (**k**) Total number of *TSC1/TSC2* mutations of each class for each lesion type. (**l**) Combinations of mutations in *TSC1* and *TSC2* mutant tumours. (**m,n**) Relative TSC2 (**m**) and TSC1 (**n**) mRNA expression in non-TSC tissues ($n = 11$) and TSC lesions grouped by mutational status (*TSC2* mutation: 1 ($n = 11$) or 2 ($n = 14$); *TSC1* mutation: 1 ($n = 2$) or 2 ($n = 3$)). FDR-adjusted *P* values from individual Welch's *t*-tests: **$P < 0.01$; *$P < 0.05$. Open circles represent individual tumours and bars represent mean and s.d. CN-LOH, copy-neutral loss-of-heterozygosity; CPM, counts per million; DEL, deletion; FS, frameshift; IFD, in-frame deletion; MS, missense; Mut, mutation; NMI, no mutation identified; NS, nonsense; SPL, splicing.

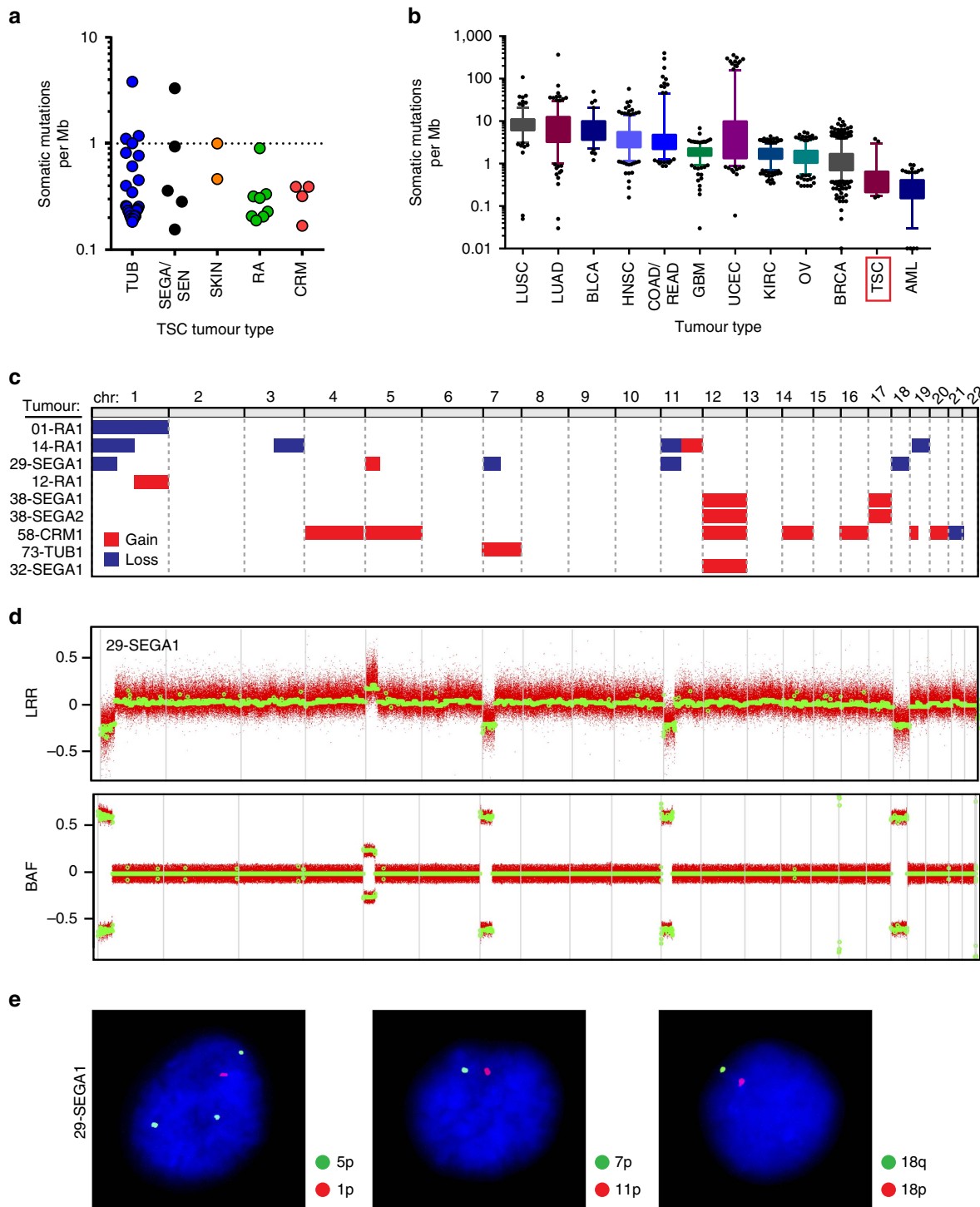

**Figure 2 | TSC lesions are infrequently mutated with large chromosome aberrations identified in a subset.** (**a**) Each symbol represents the number of somatic mutations, including point and INDELs, per Mb of genomic DNA. (**b**) Somatic mutations per Mb for cancerous and TSC lesions shown as boxes (25th to 75th percentile) and whiskers (5th to 95th percentile). Outliers shown as individual data points. (**c**) GISTIC was used to call whole and arm-level CNAs using processed SNP array data. Gains (3*n*) are shown in red and losses (1*n*) are shown in blue. (**d**) All CNAs were visually confirmed with genome-wide LRR and BAF plots. 29-SEGA1 is shown as an example. (**e**) FISH was used to confirm large CNAs. Representative nuclei, stained with DAPI in blue, show single fluorescent puncta using probes to chromosome 1p, 11p, 7p, 18p and 18q, and three fluorescent puncta using a probe to chromosome 5p, confirming losses and gains, respectively. AML, acute myeloid leukemia; BLCA, bladder urothelial carcinoma; BRCA, breast invasive carcinoma; COAD, colon adenocarcinoma; GBM, glioblastoma multiforme; HNSC, head and neck squamous cell carcinoma; KIRC, kidney renal clear cell carcinoma; LUAD, lung adenocarcinoma; LUSC, lung squamous cell carcinoma; OV, ovarian serous cystadenocarcinoma; READ, rectum adenocarcinoma; TSC, tuberous sclerosis complex; UCEC, uterine corpus endometrial carcinoma.

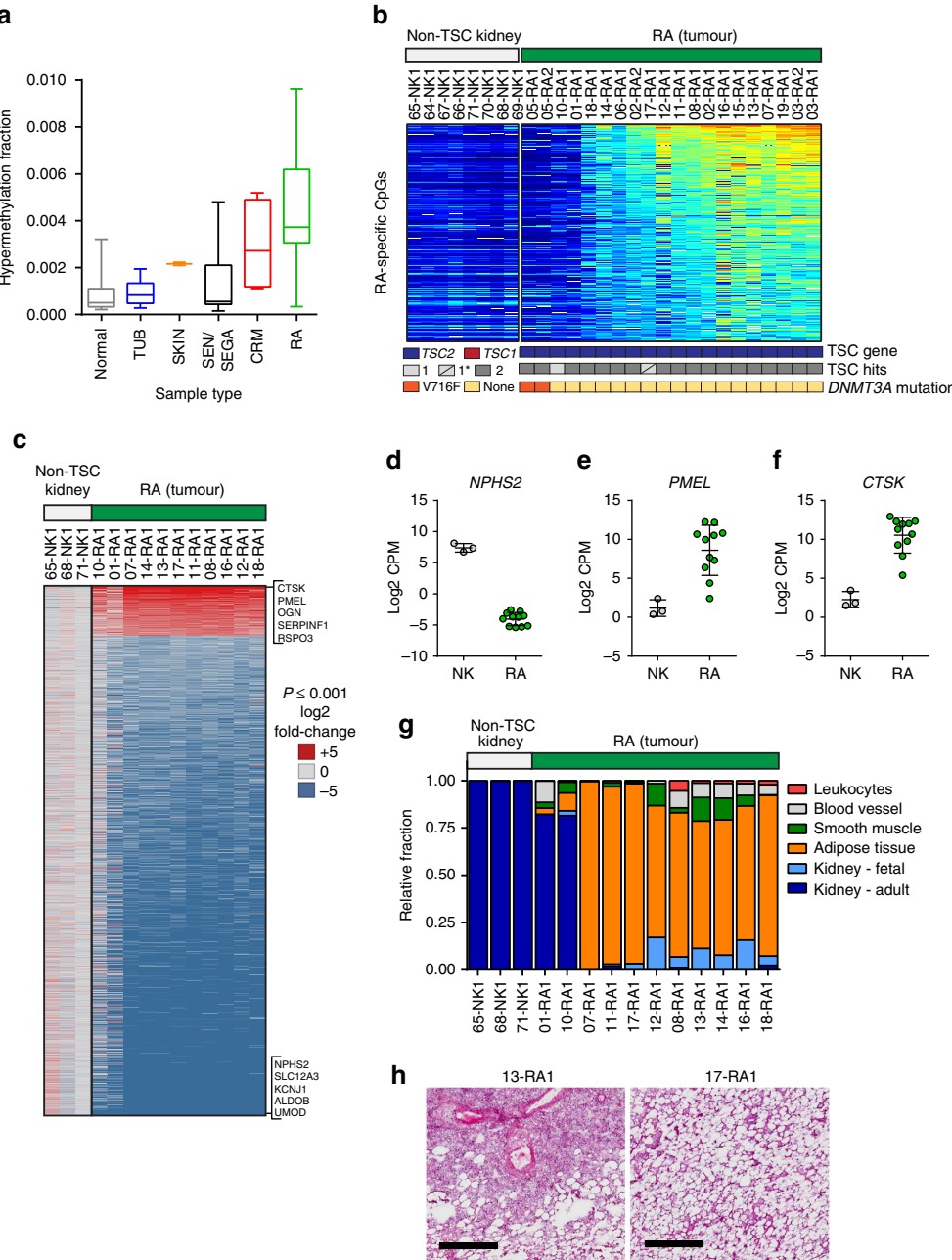

**Figure 3 | Shared methylation and transcriptional features of angiomyolipomas. (a)** The hypermethylation fraction of TSC tumours (boxes: 25th to 75th percentile; whiskers: minimum to maximum; line: median). **(b)** 240 RA-specific probes. Probe signal intensity shown from low (blue) to high (red). Samples annotated for germline mutation (blue: *TSC2*; red: *TSC1*), 1 or 2 TSC hits (germline + somatic mutation) with '1*' indicating only 1 mutation found but data from targeted *TSC1/TSC2* sequencing not available, and *DNMT3A* status (yellow: wild-type; orange: V716F somatic mutation). **(c)** Genes differentially expressed in RA tumours compared to non-TSC kidney samples (log2 fold-change +/− >2; adjusted *P*<0.001) coloured by log2 fold-changes according to the scale (blue: low; red: high). Top 5 upregulated and downregulated DEGs are indicated. **(d–f)** Normalized log2 CPM values for NPHS2, PMEL and CTSK for non-TSC kidney (NK; grey) and RA (green). Symbols represent single samples. Line: mean; error bars: s.d. (*n* = 3 non-TSC kidney; *n* = 11 RA). **(g)** The relative fraction of cell types estimated by CIBERSORT (red: leukocytes; grey: blood vessel; green: smooth muscle; orange: adipose; light blue: fetal kidney; dark blue: adult kidney). **(h)** H&E-stained tissue sections of 13-RA (showing smooth muscle surrounding vasculature and pockets of adipose tissue) and 17-RA1 (nearly exclusive adipose tissue). Scale bars, 500 μM.

(Supplementary Fig. 4a), we identified 240 CpG probes, mapping to 149 genes, with enriched methylation in RAs compared to non-TSC normal kidneys (Fig. 3b and Supplementary Data 6). Several methylated genes—including *WT1*, *SIX2*, *SLIT2*, *EMX2* and *OSR1*—are known to play roles in kidney development[35,36]. To determine whether this methylation is associated with differences (that is, decreases) in gene expression, we cross-referenced them with relative transcript levels determined

by RNAseq. Of the 127 methylated genes detected in our RNAseq assay, 13 were significantly differentially expressed in RAs, with all but one specifically showing reduced expression in tumours (Supplementary Fig. 4b). While we did not detect a statistically significant association between hypermethylation and differential expression across all genes ($\chi^2(1, n = 16,408) = 1.873, P = 0.17$), the decreased expression of 12 of 13 (92%) genes both hypermethylated and differentially expressed in RAs is

**Table 1 | Network analysis of top DEGs in RAs.**

| GO biological process* | Total genes† | Matched genes‡ | Adj. P value§ |
|---|---|---|---|
| *Decreased expression* | | | |
| Transmembrane transport | 769 | 50 | $3.74 \times 10^{-18}$ |
| Anion transmembrane transport | 61 | 11 | $3.49 \times 10^{-9}$ |
| Ion transport | 321 | 22 | $5.80 \times 10^{-9}$ |
| Sodium ion transport | 104 | 14 | $1.06 \times 10^{-9}$ |
| Sodium ion transmembrane transport | 89 | 13 | $1.61 \times 10^{-9}$ |
| Excretion | 40 | 12 | $2.13 \times 10^{-12}$ |
| Small molecule metabolic process | 1,551 | 54 | $6.41 \times 10^{-9}$ |
| Xenobiotic metabolic process | 176 | 15 | $1.08 \times 10^{-7}$ |

*Most significantly enriched GO biological processes (FDR-adjusted (adj.) $P < 0.0001$) from up to 300 DEGs (log2 fold-changes $+/- > 2$; adj. $P < 0.001$). Only those processes among the top 10 with >10 matched genes are shown. Nesting is according to GO ontology parent–child relationships.
†Total number of genes in the GO biological process.
‡Number of genes in the GO biological process that are DEGs.
§P values adjusted for multiple comparisons.

consistent with methylation-induced silencing. Four RAs lacked the methylation signature shared by the bulk of RAs, two of which (from patient 05) may be attributed to a somatic *DNMT3A*-V716F mutation predicted to affect methyltransferase activity (Fig. 3b)[37].

Next, we interrogated the RA transcriptome to establish whether gene expression patterns could provide insight to their development. For this effort, we used RNAseq data from a panel of non-TSC normal kidneys and 11 RA samples to identify 1,395 differentially expressed genes (DEGs; defined by log2 fold-change $+/- > 2$ and *limma* moderated $t$ statistic FDR-adjusted $P < 0.001$) (Fig. 3c and Supplementary Data 7)[38]. Genes with the most substantial decrease in expression included those with roles in normal kidney function, such as *NPHS2* (Fig. 3d), reflecting the loss of normal kidney tissue. Consistently, the top significantly enriched biological processes among genes decreased in RAs were primarily related to normal kidney development and function (Table 1). RAs are classified as PEComas, tumours arising from perivascular epithelioid cells (PECs) that co-express markers of melanocytes, bone, cartilage and smooth muscle, likely reflecting a neural crest origin[39]. Consistent with this, the two most highly expressed RA genes were *CTSK*, which has been proposed as a robust PEComa biomarker, and *PMEL*, which encodes a melanocyte-specific premelanosome protein (Fig. 3e,f)[40]. In fact, *PMEL* encodes the protein target of HMB-45 (gp100), a diagnostic antibody used to identify RAs and other PEComas clinically[41].

To estimate the relative proportion of different cell types in RAs, we employed CIBERSORT, a computational framework for virtually sorting complex cell mixtures using gene expression data[42]. We created a custom gene signature differentiating cell types we suspected comprise RAs: (a) adipose tissue, smooth muscle and blood vessel, which histologically define RAs; (b) adult and fetal kidney, with the hypothesis that RAs may bear more resemblance to fetal than adult kidney; and (c) leukocytes, which frequently infiltrate tumour microenvironments. As expected, CIBERSORT predicted non-TSC kidney samples to be comprised exclusively of normal adult kidney and similarly, the two RAs with DEG signatures least similar to the other RAs (01-RA1 and 10-RA1) were also predicted to contain a significant amount of normal tissue (Fig. 3g). The remainder of RAs resembled mixtures of adipose tissue, smooth muscle, blood vessels, leukocytes and fetal kidney tissue, with most showing a striking enrichment in adipose tissue (Fig. 3g). The loss of normal kidney tissue and the presence of the three known RA components, including the lipoma-like phenotype of many, were supported by histology (Fig. 3h).

**Brain lesions show evidence of significant neuroinflammation.** Analogous to the approach we took for RAs, we next performed differential gene expression analysis of the two main classes of TSC-associated brain lesions, cortical tuber ($n = 15$) and SEN/SEGA ($n = 15$), using normal non-TSC brain tissues as negative controls. We identified 3,692 DEGs (log2 fold-change $+/- > 2$; *limma* moderated $t$ statistic FDR-adjusted $P < 0.001$) in SEN/SEGAs and 297 DEGs in cortical tubers (Fig. 4a,b and Supplementary Datas 8 and 9). Almost all genes with decreased expression in cortical tubers were also decreased in SEN/SEGAs, with both lesion types showing decreased expression of genes related to synaptic transmission (Fig. 4c and Table 2). SEN/SEGAs showed a large number of uniquely decreased genes, which were associated with other normal nervous system processes (Table 2).

We found that genes most significantly increased in expression in both brain lesion types were related to the immune system and inflammation (Table 2). Antigen processing and presentation—specifically, major histocompatibility (MHC) class II—was a major process significantly enriched among increased SEN/SEGA DEGs (Table 2), which we highlighted by colour-coding a molecular map of this network according to average fold-changes in SEN/SEGAs (Fig. 4d). Most genes increased in expression in tubers were also increased in expression in SEN/SEGA, reflecting this shared immune signature (Fig. 4c). To identify genes with discordant expression between the two brain lesion types, we performed a final differential gene expression analysis between cortical tuber and SEN/SEGA. While genes uniquely decreased in expression in SEN/SEGA again mapped to normal nervous system processes, we found receptor-mediated endocytosis and angiogenesis were enriched among genes increased in SEN/SEGA compared to cortical tuber (Supplementary Table 2). This angiogenesis signature may contribute to the known vascularized nature of SEGAs. It is worth noting that while MTOR-related signalling was not identified as enriched in this analysis, we were able to detect an enrichment of MTORC1 networks in SEN/SEGA (but not TUB or RA) using a second pathway enrichment analysis (MetaCore) with reduced stringency of our analysis (Supplementary Table 3).

Finally, we wanted to explore the neuroinflammation phenotype further which we began by employing CIBERSORT to estimate the proportion of cell types constituting these lesions. Cortical tubers showed a relatively equal mixture of adult neuron and astrocyte, similar to normal non-TSC brain tissue, along with a small fraction of leukocytes (~1%) (Fig. 5a). Meanwhile, SEN/SEGAs were estimated to be enriched in less differentiated

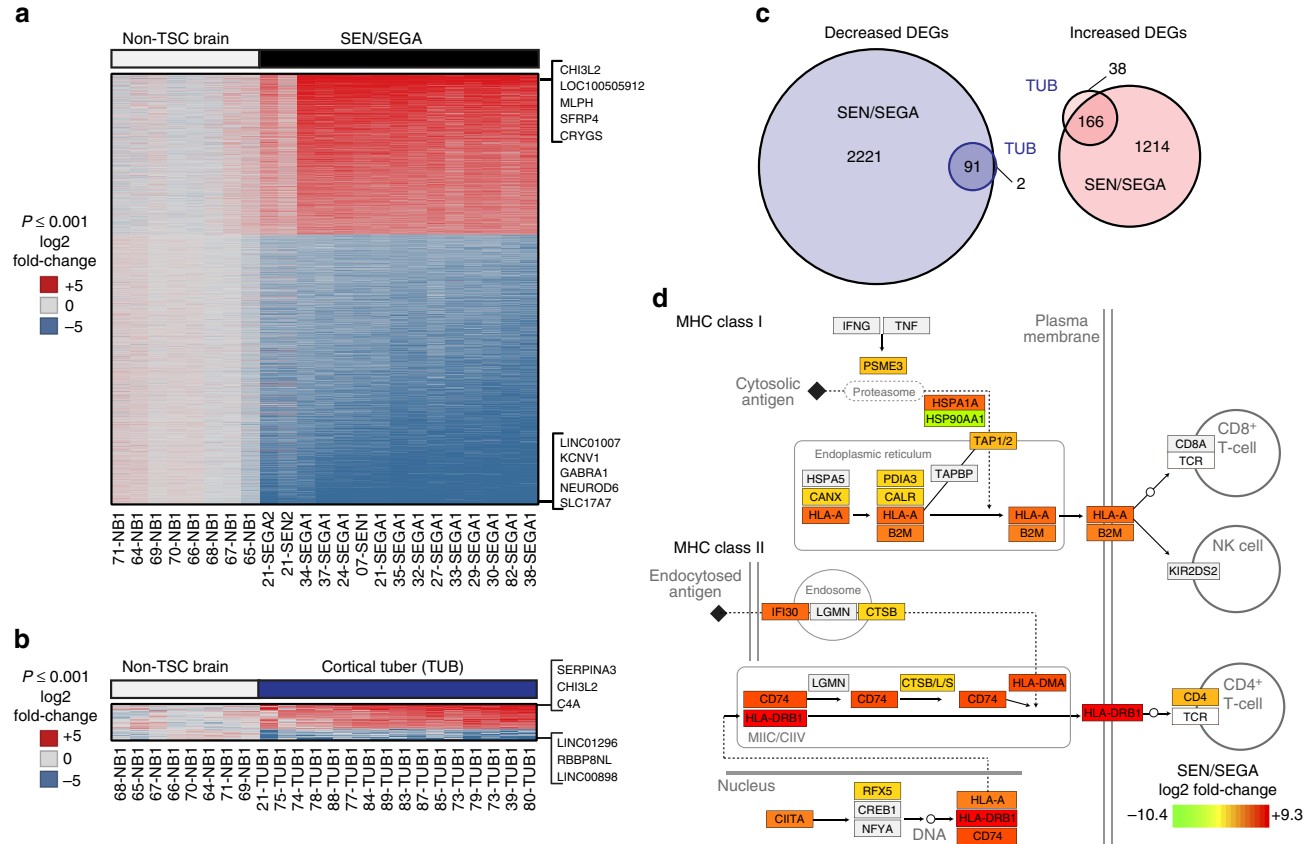

**Figure 4 | Shared gene expression features of TSC-associated brain lesions.** (**a,b**) Genes differentially expressed in SEN/SEGA (**a**) or cortical tubers (TUB) (**b**) compared to non-TSC brain (NB) samples (log2 fold-change $+/- >2$; adjusted $P < 0.001$) are coloured by log2-transformed fold-changes according to the scale (blue: low; red: high). Top 5 and top 3 increased and decreased DEGs are indicated for SEGAs (**a**) and cortical tubers (**b**), respectively. (**c**) Venn diagrams showing similarity in DEGs between TUB and SEGA (blue: decreased DEGs; red: increased DEGs). Circles are sized according to DEG number. (**d**) Molecular map of antigen processing and presentation modified from KEGG ID hsa04612. Genes are coloured according to log2 fold-changes in SEN/SEGA according to the legend.

neurons and astrocytes, a result substantiated by the decreased expression of known neuronal differentiation markers (Fig. 5b). In addition, SEN/SEGAs showed evidence of substantial leukocyte levels (12.3% mean) (Fig. 5a). To predict the relative abundance of individual immune cell components found in the leukocyte fraction of SEN/SEGA samples, we utilized a gene signature distinguishing 22 immune cell types[42]. We identified three immune cell types with relative fractions differing more than threefold between non-TSC brain and SEN/SEGAs (two-tailed Student's *t*-tests; FDR-adjusted $P < 0.05$) (Supplementary Data 10). When compared to normal brain tissue, SEN/SEGAs were predicted to be enriched in monocytes, and harbour a population of macrophages switched from a resting (M0) an activated (M2) state (Fig. 5c). Using immunohistochemistry (IHC) on sections from a fresh-frozen patient-derived SEGA, we confirmed the presence of this activated macrophage (microglia) population using CD68 (macrophage marker) and HLA-DR (a MHC-class II antigen), as well as AIF1/IBA1 (microglia marker) (Fig. 5d).

## Discussion

We have presented the most complete molecular portrait of TSC to date, adding genomic information beyond the well-described *TSC1* and *TSC2* loci. The genomes of TSC-associated lesions are relatively simple, with somatic mutation rates lower than most malignant tumours. The mutational burden of TSC lesions suggests a low mitotic index, consistent with their slow-growing nature and lack of

exposure to genotoxic therapies. Instead, the most remarkable DNA feature of TSC genomes was whole or arm-level chromosome gains and losses, which were observed in four different tumour types from just over 10% of patients in our study. These generally (seven of nine tumours) co-occurred with large aberrations to *TSC1*/*TSC2* (large deletions or CN-LOH), suggesting certain genomes may be less structurally stable. Future studies will be required to establish the role of these CNAs in TSC tumour growth.

By integrating targeted-deep sequencing that spanned introns and exons with high-resolution SNP arrays, we were able to identify pathogenic *TSC1*/*TSC2* mutations in almost 94% of patients, leaving just two classified as NMI. The remaining cases may be explained by (a) mosaicism, in which only a portion of cells (and therefore, DNA) is affected by a mutation; (b) a third TSC locus (*TSC3*); or (c) *TSC1*/*TSC2* mutations that have not yet been attributed pathogenicity, for example, intronic mutations that may affect splicing. The latter of these seems most plausible as our integrated platforms proved sensitive at detecting low-frequency mutations, including low-level mosaicism, and our WES analysis failed to provide evidence for *TSC3*. Moreover, we did identify rare, intronic *TSC2* mutations of unknown significance in one of the NMI patients. Genetic testing of biological parents and biochemical evaluation of these mutants will resolve whether one of these variants is indeed pathogenic. Overall, our data is consistent with a recent thorough evaluation of 53 NMI patients by targeted-deep sequencing that concluded 85% of cases could be explained by low-frequency mosaic mutations or mutations in introns[17].

**Table 2 | Network analysis of top DEGs in TSC-associated brain lesions.**

| GO biological process* | Total genes[†] | SEN/SEGA | | TUB | |
|---|---|---|---|---|---|
| | | Matched genes[‡] | Adj. P value[§] | Matched genes[‡] | Adj. P value[§] |
| *Increased expression* | | | | | |
| Immune response | 408 | 41 | $1.69 \times 10^{-21}$ | 20 | $5.27 \times 10^{-9}$ |
| Regulation of immune response | 146 | — | — | 12 | $3.15 \times 10^{-8}$ |
| Innate immune response | 869 | — | — | 33 | $3.01 \times 10^{-11}$ |
| Interferon-gamma-mediated signalling pathway | 77 | 13 | $2.88 \times 10^{-10}$ | — | — |
| Complement activation | 51 | 11 | $5.56 \times 10^{-10}$ | — | — |
| Complement activation, classical pathway | 64 | 12 | $4.37 \times 10^{-10}$ | — | — |
| Antigen processing and presentation | 62 | 16 | $4.33 \times 10^{-15}$ | — | — |
| Antigen processing and presentation of exogenous peptide antigen via MHC-class II | 93 | 12 | $2.64 \times 10^{-8}$ | — | — |
| Inflammatory response | 360 | 22 | $4.29 \times 10^{-8}$ | 17 | $1.32 \times 10^{-7}$ |
| Extracellular matrix organization | 316 | 20 | $1.03 \times 10^{-7}$ | — | — |
| Cytokine-mediated signalling pathway | 279 | — | — | 14 | $8.67 \times 10^{-7}$ |
| | | | | | |
| *Decreased expression* | | | | | |
| Synaptic transmission | 432 | 59 | $6.86 \times 10^{-38}$ | 10 | $1.58 \times 10^{-5}$ |
| Synaptic vesicle exocytosis | 50 | 13 | $1.50 \times 10^{-12}$ | — | — |
| Neurotransmitter transport | 36 | 10 | $3.10 \times 10^{-10}$ | — | — |
| Neurotransmitter secretion | 65 | 14 | $2.48 \times 10^{-12}$ | — | — |
| Calcium ion-dependent exocytosis of neurotransmitter | 33 | 11 | $5.87 \times 10^{-12}$ | — | — |
| Regulation of membrane potential | 127 | 19 | $1.75 \times 10^{-13}$ | — | — |
| Ion transport | 321 | 29 | $2.54 \times 10^{-14}$ | — | — |
| Regulation of calcium ion-dependent exocytosis | 31 | 10 | $7.43 \times 10^{-11}$ | — | — |
| Neurological system process | 52 | 11 | $6.79 \times 10^{-10}$ | — | — |

*Most significantly enriched GO biological processes (FDR-adjusted (adj.) $P < 0.0001$ from up to 300 DEGs (log2 fold-changes $+/- > 2$; adj. $P < 0.001$). Only those processes among the top 10 with $>10$ matched genes are shown. Nesting is according to GO ontology parent–child relationships.
†Total number of genes in the GO biological process.
‡Number of genes in the GO biological process that are DEGs. '—' indicates no significant enrichment.
§P values adjusted for multiple comparisons. '—' indicates no significant enrichment.

In addition to finding germline mutations, we also provided a detailed description of the second hit landscape across TSC tumours. Our observation of widespread somatic TSC inactivation in RA and less common second hits in cortical tubers is consistent with previous studies[27–32,43]. Epigenetic silencing of TSC1/TSC2 has been postulated to explain a portion of 1-hit tumours and in fact, there has been some evidence that TSC1/TSC2 are subject to methylation[44,45]. However, we found no evidence of promoter methylation in 63 TSC-associated tissues analysed, reducing the likelihood that this mechanism contributes significantly to TSC1/TSC2 inactivation in TSC.

Although our second hit rate in cortical tubers (35%) was higher than most previous studies, somatic TSC inactivation in cortical tubers is clearly a less frequent and more sporadic event. This suggests that either only a small portion of the tuber is affected by a second hit (for example, one cellular component, such as giant cells), hindering its identification or that monoallelic inactivation of TSC1/TSC2 is sufficient for cortical malformation. The latter is supported by the fact that cortical tubers form prenatally, are found in a majority of TSC patients, and despite some evidence of proliferation[46], lack appreciable growth in size or number over time. These features are consistent with a developmental origin rather than neoplastic formation via the sporadic acquisition of somatic TSC1/TSC2 mutations over time. This concept of haploinsufficiency is consistent with other features of this disease, such as cognitive and behavioural impairments, and to some degree, epilepsy[47–49]. The second hits found in a minority of cortical tubers may contribute to tuber pathology, although they are unlikely to represent a requirement for their formation.

Interestingly, two of the somatic TSC mutations in cortical tubers (44-TUB1 and 50-TUB1) were unusual and appeared to involve the loss of TSC1 or TSC2 introns. The deleted introns were continuous and breakpoints appeared to be precisely at exon–intron boundaries, raising the possibility that they are the consequence of somatic retroduplication events where reverse-transcribed copies of genes lacking introns are integrated into the genome, forming a processed pseudogene. Such events are widely present in human germline evolution but also have recently been reported to occur in cancers[50–52]. An added layer of intrigue stems from the fact that one of these events affected TSC1 but was found in a patient with a pathogenic germline TSC2 mutation. A similar case was previously reported in which a low-frequency somatic TSC1 mutation was identified in the periungual fibroma from a mosaic TSC2 patient[53]. While the authors suggested it was unlikely that a monoallelic mutation in TSC1 could cooperate with a germline TSC2 mutation to drive MTOR activation, we feel that together, our reports with similar observations from unique patients supports the idea that trans-heterozygous TSC1/TSC2 mutations may contribute to tumorigenesis in TSC. Interestingly, $Tsc1+/-;Tsc2+/-$ compound heterozygous mice show increased numbers of hippocampal GFAP-positive astrocytes compared to either single heterozygote mice, suggesting potential epistatic interaction between monoallelic mutation of the two genes[54].

Despite generally stable DNA genomes, TSC lesions were defined by marked and uniform changes in gene expression. In fact, the DEGs we identified in SEN/SEGAs constitute nearly 20% of all genes identified in our RNAseq assay. These RNA signatures were shared by tumours regardless of TSC1/TSC2 mutational status or presence of second hits. While enriched MTORC1 signalling was observed in SEN/SEGA, it was only significant when the pathway analysis stringency was reduced and it was not detected in RA or TUB. This result may be a

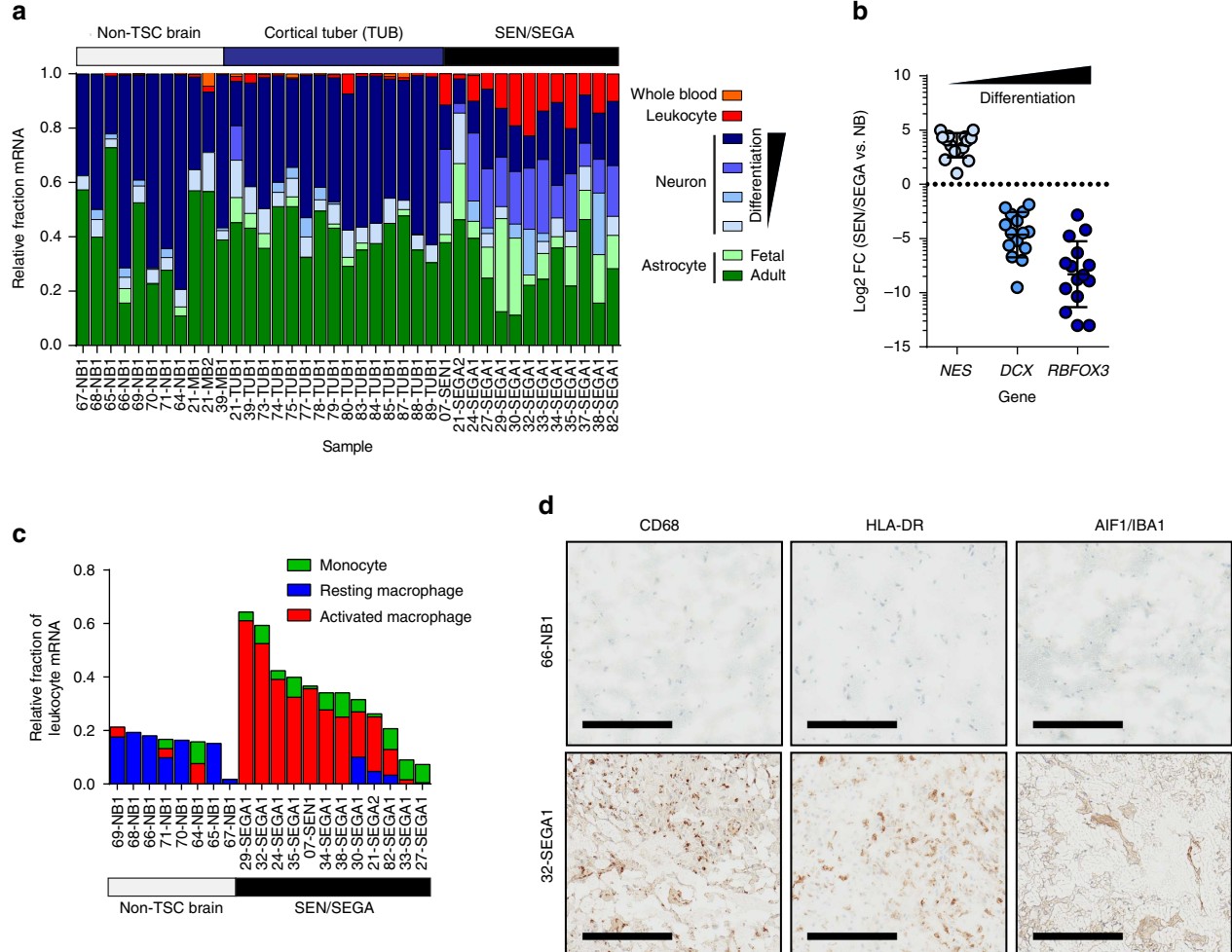

**Figure 5 | Immune response is observed in TSC brain lesions.** (**a**) Relative fraction of cell types estimated by CIBERSORT ($P < 0.05$). Orange: whole blood; red: leukocytes; dark blue: differentiated neuron; light blue: undifferentiated neuron; light green: fetal astrocyte; dark green: adult astrocyte. (**b**) Fold-change in transcript abundance in SEN/SEGA compared to non-TSC brains for three markers of neuronal differentiation. Lines: means; error bars: s.d. (**c**) Relative fractions of 22 immune cell types estimated by CIBERSORT. Those significantly enriched in SEN/SEGA compared to NB (FDR-adjusted $P < 0.05$) are displayed as their relative fraction of leukocyte RNA per sample. (**d**) Fresh-frozen tissue sections from NB and SEGA were immunostained for CD68 (macrophages), HLA-DR (MHC-Class II) and AIF1/IBA1 (microglia). 66-NB1 and 32-SEGA1 are shown. Scale bars, 200 µM.

consequence of tumour heterogeneity (that is, if only a portion of a tumour or specific cells—such as giant cells—bear second hits and are strongly driven by MTORC1 signalling) and is also consistent with the molecular role that MTOR regulation plays in translational control (versus transcription)[55]. Moreover, this dramatic expression signature also suggests that a strong (and common) cell-of-origin expression signature may be dominating over additional molecular signalling changes. This is best supported by the RA expression signature, which showed features of several cell types known to be derived from the neural crest, the proposed cell-of-origin for this lesion[56]. In addition, the SEN/SEGA gene expression signature showed evidence of less differentiated neurons and glia, consistent with their proposed derivation from neural stem progenitor cells (NSPCs), early and shared precursors of both of these cell types[57,58].

Throughout this study, we employed CIBERSORT to generate computational estimates for the relative proportion of cell types in TSC lesions using gene expression data. While the original report of this methodology used microarray data and focused on immune cell types, we have extended its use to RNAseq data and a host of additional cell types. It is worth noting that CIBERSORT

can only generate predictions using input cell types; therefore, additional cellular components of these tumours beyond those we tested may exist. We used these molecular tools to detect neuroinflammation associated with TSC brain lesions. Inflammation has been previously documented in TSC lesions, both in the brains of TSC animal models and in patient tissues[59–62]. Work from Zhang et al.[62] suggested that this inflammation is directly related to hyperactive MTORC1 signalling and is not merely a result of seizure activity. Inflammation has even been detected in prenatal TSC brain lesions, suggesting it is an early, and sustained, feature of TSC pathology[63]. In our study, we uncovered inflammation in both cortical tubers and SEN/SEGAs, although the extent of inflammation appeared much more substantial in SEN/SEGAs. In fact, computational sorting of RNAseq data estimated as much as 20% of the mRNA fraction of SEN/SEGA tissues was associated with leukocytes, with a specific enrichment of activated macrophages. We postulate that this macrophage signature largely reflects the activation of brain-resident microglia, the primary immune cell component of the central nervous system (CNS) and known mediator of neuroinflammation, which is supported by positive AIF1/IBA1

staining in SEGA tissue. Reactive astrocytes, known key mediators of innate immunity in the CNS and neuroinflammation, also likely contribute to the immune signature we detected. While inflammation may serve a protective role in response to acute brain injury, triggering angiogenesis and promoting tissue repair, chronic neuroinflammation may instead be destructive and contribute to neuronal damage, as is the case in CNS pathologies like Alzheimer's and Parkinson's disease. Important future work should focus on defining the relationship between neuroinflammation and neurological symptoms of TSC, including seizure activity and cognitive impairment, as well as evaluation of anti-inflammatory agents in the treatment of TSC.

## Methods

**Patients and samples.** Samples from TSC patients or non-TSC organ donors were acquired from the NIH NeuroBioBank's Brain and Tissue Repository at the University of Maryland, Houston-McGovern Medical School at the University of Texas, Cincinnati Children's Hospital Medical Center, New York University School of Medicine and Helen DeVos Children's Hospital. All tissues used in this study were fresh-frozen and collected at the time of surgery or procedure or post-mortem (see Supplementary Data 1 for details). This study was approved by the Van Andel Research Institute (VARI) Institutional Review Board (IRB). Written informed consent was obtained from all human participants providing samples. Samples were reviewed by a certified clinical pathologist to confirm tissue type and assess integrity, whenever possible (samples with inconsistent, unlikely to be consistent or unclear diagnoses were excluded from the study). Samples were also excluded if they failed to produce usable data on two of three DNA platforms (WES, SNP array and targeted TSC sequencing), with the exception of one non-tumour tissue sample in which the germline mutation was identified in the completed platform (eliminating the need for the additional platforms to be completed).

**Immunohistochemistry.** For immunohistochemistry, 5 μm fresh-frozen tissue sections were fixed and stained with primary antibodies (CD68: 1:100; HLA-DR: 1:40; AIF1/IBA1: 1:500), secondary antibodies (Ultramap anti-mouse HRP multimer) and detection reagent (Ventana Chromomap DAB). Slides were processed on the Discovery Ultra platform (Ventana) and imaged using the ScanScope XT digital pathology slide scanner (Aperio).

**DNA and RNA isolation.** The specific method for DNA and RNA isolations is indicated in Supplementary Data 1. For majority of frozen tissues, DNA and RNA was simultaneously isolated using a modified version of the method described in Pena-Llopis and Brugarolas[64]. Briefly, tissues were lysed and homogenized using mirVana kit lysis buffer (Ambion), a micropestle and QIAshredder columns (Qiagen). DNA was isolated using AllPrep columns (Qiagen) while flow-throughs were used to isolate RNA using an acid phenol–chloroform extraction and the mirVana kit (Ambion). DNA integrity was confirmed by agarose gel electrophoresis and RNA integrity was confirmed using a BioAnalyzer 2100 (Agilent). DNA and RNA concentrations were determined using a Qubit 2.0 fluorometer (Invitrogen).

**Whole-exome sequencing.** DNA sequencing was completed at the HudsonAlpha Institute for Biotechnology (HAIB) Genomic Services Laboratory (GSL) or Beijing Genomics Institute (BGI) at the Philadelphia Children's Hospital. Briefly, exonic DNA was enriched using a SeqCap EZ Human Exome Library v3.0 (NimbleGen) or SureSelect Human All Exon capture kit (Agilent) from genomic DNA. Libraries were pooled and clustered at 16–18 pM on the HiSeq 2500 or HiSeq 2000 with high output flowcells and sequenced at 100PE according to Illumina protocols. Fastq files were generated using Illumina software, aligned to the hg19 genome with BWA-MEM and variants called using Haplotype Caller in GATK. Filtered variants were annotated with Variant Effect Predictor (VEP) and imported to GEMINI. Detailed methods can be found in Supplementary Methods.

**Statistical methods.** For the TSC1/TSC2 expression analysis, pair-wise Welch's t-tests (in GraphPad Prism 6 for Windows, version 6.07) of 5 groups of data (for Fig. 1m: non-TSC tissue; NMI and TSC1 tumours; 0, 1 or 2 truncating TSC2 mutations; for Fig. 1n: non-TSC tissue; NMI and TSC2 tumours; 0, 1 or 2 truncating TSC1 mutations) were followed by false discovery rate (FDR) correction (in R) to generate corrected P values. This approach was taken because samples failed Bartlett's test for homogeneity of variances, ruling out ANOVA as an option. Truncating mutations included nonsense, frameshift, splicing and large deletions. Tumours with truncating germline mutations and CN-LOH were classified as harbouring two truncating mutations (because CN-LOH duplicates the germline mutant allele). As a priority for visualization, only non-TSC tissue and 1 or 2 truncating mutation groups were shown in Fig. 1m,n although all were included in

the statistical analysis. For immune cell type analysis by CIBERSORT (Fig. 5c), the relative fraction of each cell type in SEN/SEGA was divided by the fraction in non-TSC brain. Individual two-tailed student's t-test P values were adjusted via the FDR method using R. Those cell types with $+/->$ threefold changes and FDR-adjusted $P < 0.05$ were included in the panel. To test the association between hypermethylation and differential expression, we identified genes covered by both HM450 and our RNAseq assay and categorized each as being hypermethylated and a DEG (13) or not a DEG (114), or not hypermethylated and a DEG (1,156) or not a DEG (15,125). These values were entered into a $2 \times 2$ contingency table and a $\chi^2$ test performed in GraphPad Prism 6.

**Targeted TSC1/TSC2 sequencing.** We designed a custom targeted enrichment kit (SeqCap EZ Choice Library, NimbleGen) with comprehensive coverage of TSC1 and TSC2, including upstream and downstream elements (including PKD1), exons and introns. Samples were multiplexed (9–10 per library hybridization) and sequenced similar to above at the HAIB GSL using Illumina reagents and the HiSeq 2500. Alignment and variant calling and annotation were performed similar to WES. In addition, we explored mutations present at low allele frequencies down to 0.5% in the deep sequencing experiment by recalling mutations using LoFreq[65] and VarDict[66]. Detailed information can be found in Supplementary Methods.

**TSC1/TSC2 mutation calling.** To be included in Supplementary Data 2, TSC1/TSC2 variants ($> 10 \times$ total read-depth) were required to be either published in the tuberous sclerosis Leiden Open Variation Database (LOVD; www.LOVD.nl/TSC2; www.LOVD.nl/TSC1) (v2.0 Build 36) as pathogenic or probably pathogenic, or if not present in LOVD (or 'unknown' pathogenicity in LOVD), determined to be rare (not present in 1000 genomes database) and impactful to gene function (medium/moderate or high impact SNV or INDEL). All large deletions and CN-LOH events affecting TSC1 and TSC2 were assumed detrimental to gene function and included. For non-tumour (normal) tissues ($n = 33$) or tumours that were paired with non-tumour tissue from the same patient ($n = 42$), the germline or somatic origin of mutations could be absolutely determined. We then used information from these samples to establish features of germline and somatically derived mutations to predict the origin of mutations in unpaired tumours ($n = 36$), described in detail in Supplementary Methods.

**Somatic mutation analysis.** Somatic SNVs were identified using MuTect using default settings and annotated using VEP and GEMINI[67]. INDELs were detected and characterized in both tumour and matched normal samples using Pindel[68]. To call a somatic INDEL, we required $> 5 \times$ coverage in both tumour and normal samples, $> 5$ reads supporting the variant allele in the tumour with 0 reads in the matched normal sample, and a variant allelic fraction of $> 0.10$ in the tumour sample. Somatic mutation rates were determined by normalizing the combined number of somatic SNVs and INDELs by the total number of bases with $> 5 \times$ read-depth in both tumour and normal samples. We excluded reads with base quality $< 20$ at each mutation locus. The mutation rates for cancers presented in Fig. 2b were obtained from Kandoth et al.[34]. Supplementary Data 3 includes only SNVs and INDELs passing more stringent criteria: SNVs required $> 10 \times$ read-depth at the variant position and 0 variant reads in the normal samples; all somatic INDELs were manually inspected in the Integrative Genomics Viewer (Broad Institute) and clear artifacts were excluded.

**RNA sequencing and differential gene expression analysis.** RNA sequencing was completed at the HAIB GSL. Briefly, messenger RNA (mRNA) libraries were prepared using NEBNext reagents (New England BioLabs), and samples underwent directional sequencing on the Illumina HiSeq 2500 using 100 bp paired end reads. Quality-filtered reads were aligned to the hg19 genome using Subread. Raw read counts obtained using FeatureCounts were imported into R for differential expression analysis via limma[38] and counts per million (CPM) calculated and log2-transformed using voom[69] followed by trimmed mean of M-values (TMM) normalization. GeneAnalytics (LifeMap Sciences; geneanalytics.genecards.org) was used for primary gene set enrichment analysis[70]. A maximum of 300 gene symbols were used and up to 10 GO biological processes with medium or high matching scores (FDR-adjusted $P < 0.05$) were included in the results. Only processes with at least 10 matched genes were shown in Tables 1 and 2. A follow-up enrichment analysis to search for MTORC1-related signatures was completed using MetaCore. For this, gene-level fold changes and adjusted P values were imported into MetaCore version 6.29 build 68613 (Thomson Reuters) for pathway analysis. Pathway analysis was performed using the Pathway Maps One-Click Analysis on genes with an absolute log-fold change $> 1$ and FDR-adjusted $P$-value $< 0.001$. Pathway Maps with a FDR-adjusted $P$-value $< 0.05$ were considered significant. RNAseq variant calling was conducted using GATK (v3.0) using the suggested Best Practices parameters and with a two-pass STAR (v 2.4.2a) alignment method to the hg19 genome. DNA variants identified in RNAseq are indicated in Supplementary Data 2.

**CIBERSORT.** CIBERSORT[42] was used to estimate the relative fraction of cell types. Publically available RNA sequencing data was downloaded from the NCBI Short

Read Archive (http://www.ncbi.nlm.nih.gov/sra) (see Supplementary Methods for detailed information). The values for the iPSC neurons were duplicated into two columns to meet CIBERSORT input requirements. Read quality was assessed using FASTQC v. 0.11.3 (http://www.bioinformatics.bbsrc.ac.uk/projects/fastqc/). Reads were aligned to the hg19 genome using Subread (v1.4.5) with default parameters. Raw read counts were obtained as described for RNAseq. For immune cell types, the LM22 gene signature was used[41]. Only samples with estimates yielding $P$ values < 0.05 were reported.

**SNP arrays and copy number analysis.** Copy number analysis was performed using Infinium HumanOmni2.5S Arrays (Illumina) at the HAIB GSL. Briefly, genotypes were called and total copy number, log-R ratio (LRR) and B-allele frequency (BAF) estimated for each SNP using IDAT files in GenomeStudio (v2011.1, Illumina). Total genome-wide copy number estimates were refined using tangent normalization and individual copy number estimates underwent segmentation per-sample arm-level and gene-level copy ratios were identified from segmented data using GISTIC. Purity and ploidy estimates and allelic integer copy number (including regions of copy-neutral loss-of-heterozygosity) were calculated from LRRs and BAFs using ASCAT. Arm-level copy number events determined by GISTIC 2.0 were visually validated in genome-wide LRR and BAF plots generated by ASCAT 2.4. Chromosomes 9 and 16, as well as the region in chromosome 9q containing *TSC1* and the region in chromosome 16p containing *TSC2*, were visually inspected using genoCN to validate loci with copy-neutral loss-of-heterozygosity and focal deletions as reported by ASCAT 2.4 and/or GISTIC. Copy number events detected only visually because of low tumour purity or low signal were also reported.

**Array-based DNA methylation assay.** DNA methylation profiling was completed using Infinium HumanMethylation450 BeadChips (Illumina) at the University of Southern California Epigenome Center to obtain DNA methylation profiles, which were analysed using the same pipeline used for The Cancer Genome Atlas (TCGA) project. Briefly, bisulfite conversion of genomic DNA was performed with the EZ-96 DNA Methylation Kit (Zymo Research). After quality control measures, bisulfite-converted DNA was whole genome amplified and fragmented prior to hybridization to BeadArrays, which were scanned using the Illumina iScan technology. IDAT files were used to extract the intensities and calculate beta values for each probe and sample with the R-based *methylumi* package. A $P$ value comparing the intensity of each probe to the background level was calculated and data points with detection $P$ values > 0.05 were deemed not significantly different from background measurements. Detailed information can be found in Supplementary Methods.

**Fluorescent *in situ* hybridization.** FISH probes were prepared from purified BAC clones (BACPAC Resource Center; bacpac.chori.org); see Supplementary Methods for specific BAC probes and detailed information. Briefly, each clone was labelled with Green-dUTP, Orange-dUTP or Red-dUTP by nick translation. Tumour touch preparations were made on glass slides, which were fixed, dried, aged, digested and washed. Slides were placed in 1% formaldehyde, washed and dehydrated in an ethanol series. Slides were then denatured, washed and air-dried. FISH probes were denatured probe was applied to each sample slide. Coverslips were adhered and slides hybridized overnight in a ThermoBrite hybridization system (Abbott Molecular). Post-hybridization, slides were washed with 2 × SSC and briefly rinsed with water. Slides were dried and counterstained with VectaShield mounting medium with 4′-6-diamidino-2-phenylindole (DAPI). Image acquisition was performed at ×600 or ×1,000 system magnification with a COOL-1300 SpectraCube camera (Applied Spectral Imaging-ASI) mounted on an Olympus BX43 microscope. Images were analysed using FISHView v7 software (ASI) and at least 200 interphase nuclei were scored for each sample.

**Data availability.** All raw data has been deposited in the Database of Genotypes and Phenotypes (dbGaP) under the accession code phs001357.v1.p1. All other remaining data are available within the Article and Supplementary Files, or available from the authors upon request.

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

## Acknowledgements

We are grateful to all the patients and families who contributed to this study. A portion of human tissue was obtained from the NIH NeuroBioBank's Brain and Tissue Repository at the University of Maryland (Baltimore, MD). We thank members of the MacKeigan laboratory and TSC Pathway of Hope External Advisory Board (Peter Laird, Peter Saltonstall and Len Post) for critical discussions and feedback. We also thank Nicole Doppel for project management; Jennifer Webb, Sarah Nota, Molly Griffith and Maxwell Mays for clinical coordination; Alejandro Salah for clinical information; and Braden Boone, Lisa Turner and Jennifer Kordich for technical assistance and experimental insights. This work was supported by grants and funding from the Michigan Strategic Fund, Van Andel Research Institute, Tuberous Sclerosis Alliance, Blue Cross Blue Shield of Michigan Foundation, Great Lakes Scrip, Rockford Construction, Colliers International, Team Hannah for TSC and individual donors. J.P.M. has research support from the NIH National Cancer Institute (R01CA197398) and the Tuberous Sclerosis Alliance. D.A.K. has research support from the NIH National Institute of Neurological Disorders and Stroke (U01-NS082320, U54-NS092090 and P20-NS080199) and the Tuberous Sclerosis Alliance.

## Author contributions

K.R.M., D.A.K. and J.P.M conceptualized the study; W.Z., M.J.B., K.R.M., J.P.M. and M.E.W. analysed DNA sequencing data, H.S. and D.J.W. analysed DNA methylation array data; J.S., M.E.W. and A.D.C. analysed copy number array data; M.J.B., K.R.M., M.E.W. and J.P.M. analysed RNAseq data, K.R.M., K.E.D-R., K.A.S., J.K. and J.P.M. performed biological experiments; S.L.C. performed histopathology; K.R.M. and J.P.M. wrote the manuscript with input from all authors; J.P.M. acquired funding for this study; O.D., S.T.D., K.S.A., H.N. and D.A.K. provided samples; M.E.W., H.N., H.S., A.D.C. and J.P.M. supervised the study.

## Additional information

**Competing interests:** The authors declare no competing financial interests.



