## [Peer review file · Nature Communications]

Reviewers' comments:

Reviewer #1 (Remarks to the Author):

The mutational spectrum of 78 TSC tumors and 33 non-tumor TSC tissues was extensively evaluated to reveal mutations in TSC1 and TSC2 and relatively few additional somatic mutations. The work is notable for the numbers of samples examined comprising several important types of tumors (mostly cortical tubers, renal angiomyolipomas, and subependymal nodules or subependymal giant cell astrocytomas) and rigorous investigation using multiple experimental and bioinformatic approaches. Similar conclusions about the low mutation rate were drawn for renal angiomyolipomas in another study (PLoS Genet. 2016;12:e1006242), but only three of these patients had TSC. Unusual findings in the current study included the loss of seven or more introns in two somatic TSC mutations and the possibility of transheterozygous TSC1/TSC2 inactivation evidenced by a somatic TSC1 mutation in a patient with a germline TSC2 mutation. Analysis of gene expression signatures for each tumor type provided insights into cellular composition and cell of origin for the different tumors.

There were several minor issues:

- 1) The justification for combining SEN/SEGAs was not clearly explained. The numbers for each of these tumor types should be more obvious in the methods. SEGA alone is used to describe some of the results and it is not clear if results from SENs were excluded to arrive at these conclusions.
- 2) It may be overreaching to conclude that all the underexpressed genes represented in Table 1 are important in normal kidney development and function. Also, please comment on the extent to which these results may reflect the admixture of different cell types in angiomyolipomas compared to normal kidney. In the table legend, ["---" indicates no significant enrichment] does not appear in this table. In tables 1 and 2, it does not appear that the numbers of biological processes listed match the purported top 10.
- 3) In Fig 1c, blue bars indicate exons but there also appear to be regularly spaced blue bars that are shorter than the longer exon bars. In Fig 1d the key for deletion in the upper left corner should be moved further out of line with from the data so it is not confused with the results for 06-MG1. Fig 1e the Y-axis label for LRR 1.0 is missing. Fig 1m and 1n is confusing as labeled on x-axis, since it includes non-TSC tissue for first mark and does not indicate TSC tumors for second two marks, and the 1 and 2 are not immediately evident that they refer to the number of truncating mutations in the tumors. It is difficult to see the lines among the blue circles in 1m and these lines are not defined in the legend. CPM should also be defined in Fig 1 legend.
- 4) In legend of Fig 2, define the tumor types abbreviated in 2b.
- 5) In legend of Fig 3, define the 1* in 3b.
- 6) In the methods, the approach to defining germline and point mutations is defined and mixed together somewhat with results. It would be interesting to know the median and range for somatic point mutation VAF, not just that they were < 40% VAF. Also in the methods, what defined positive FISH results when scoring 200 interphase nuclei?
- 7) In the discussion, the assertion that this is the first transheterozygous TSC1/TSC2 inactivation should consider similar results obtained on one tumor in a previous study (Hum Mol Genet. 2014;23:2023-9). It is not clear how MTOR pathway alterations were assessed to conclude that "Surprisingly, we did not identify MTOR signaling, or anything related to this pathway, in our tumor-specific pathway analyses..."

Reviewer #2 (Remarks to the Author):

In this study, Martin et al. collect an impressive amount of information regarding the genomic changes associated with tuberous sclerosis and transcriptional information that provides some insight into how these genomic changes lead to abnormal cell growth. This study catalogues putative first- and second-hit mutations in both normal and abnormal tissue, which provides an interesting view of how the mechanisms that lead to these mutations differs. They also perform transcriptional assessment of these tissues, which largely reaffirms what is known about the cell composition of renal angiomyolipomas (RAs) and subependymal giant cell astrocytomas (SEGAs). They also identify transcriptional changes that are likely due to inflammatory infiltrates, which they are able to confirm with immunohistochemistry. The studies are comprehensive and the analyses appear to be well performed. While the analyses showing the likely mechanisms of first- and second-hit mutations are very interesting, it is unclear to what degree the expression profiling experiments contribute to novel biological insights.

1. There are some areas that are not immediately clear regarding the mutations identified in TSC1 and TSC2. For example, the initial WES identified a majority of the mutations but it is not clearly described in the text of the results how many of these mutations were found in normal tissues versus abnormal tissues.
2. Categorizing cortical tubers as a "tumor" is misleading, and the authors note in the discussion that these lesions may be more appropriately categorized as a developmental malformation. It appears that not all of the cortical tubers had targeted deep sequencing of TSC1/TSC2, and it would be interesting to know how read depth affected the likelihood of identifying a second hit mutation in cortical tubers versus other tissues. It would be interesting to compare the allele frequencies of second hit mutations in RA/SEGAs versus cortical tubers. Are the allele frequencies of second hit mutations in cortical tubers low enough to suggest that deep sequencing is missing a significant portion of mutations that are below the alternative allele frequency detection limit? Or are alternate allele frequencies similar between RA/SEGAs and cortical tubes, suggesting that they should have been identified and that many cortical tubers can arise without a second hit mutation?
3. It would be interesting to analyze the breakdown of tissues where chromosomal number abnormalities were identified to know whether certain tissues are more or less likely to have rearrangements. For example, only one cortical tuber demonstrated a CNA, whereas several RA/SEGAs were identified, and it would be interesting to know whether cortical tubers are significantly less likely to have CNAs.
4. The methylation studies identify many genes with hypermethylated promoters, and about 10% of these genes were found to be differentially expressed with RNAseq. It would be interesting to know whether this is a significant enrichment in differentially expressed genes or could have been arrived at by chance alone.
5. The differential expression analysis states that the criteria were $\log_2 > +/- 2$. Is this really the case (i.e. requiring a fold change >4) or is it really $\log_2 > +/- 1$? In addition, the authors state that the other criteria for differential expression is a p-value less than 0.001. It is unclear whether this p-value is corrected for multiple comparisons, and if it is

not, it would be important to know what the estimated false discovery rate at that p-value is.

6. The authors use CIBERSORT to evaluate the likely cellular composition of the tissue based on the gene expression patterns that are present. As the authors point out, this method is only as good as the data for the cell types of interest. The authors identify many differentially expressed genes that are associated with inflammation, and they suggest that this is due to inflammatory infiltrates. This is shown using IHC in a SEGA, but is it possible that reactive astrocytosis could lead to some of these changes in cortical tubers without lymphocyte infiltration? This is a particularly important question since the IBA staining in Figure 5d is not very convincing.

7. It might be interesting to note if and how many of these patients received mTOR inhibitors and whether this had any appreciable effect on the gene expression patterns.

8. In the introduction, the authors state that several of the manifestations of TSC such as epilepsy and autism are due to hamartomas or cortical tubers. However, there are a number of patients with TSC who do not have tubers but do have epilepsy. Either they have lesions that are below the resolution of current imaging techniques or haploinsufficiency of TSC1/2 is sufficient to lead to epilepsy. This would be worth adding to the discussion.

9. In figure 1m, the reduction in TSC2 mRNA appears very small. How do the authors interpret this result and its clinical significance?

10. Based on Figure 2b, the authors state that "This mutation rate is substantially lower than almost all malignant tumor types, with the exception of acute myeloid leukemia (AML)." Since somatic mutation frequency increases exponentially with patient age, how do the authors' results change once you factor in the age of the patient? In other words, is the mutation rate in TSC lesions lower than that of non-TSC tumors in patients of similar age?

Response to Reviewers

We appreciate the time spent on this manuscript by our reviewers, as well as the constructive suggestions they have made to improve the significance and quality of our work. Answers to each comment presented by the reviewers are detailed below. We feel that the changes we have made in response to these comments strengthen the data that was presented before, and provide additional support for our conclusions.

Referee 1: "The work is notable for the numbers of samples examined comprising several important types of tumors (mostly cortical tubers, renal angiomyolipomas, and subependymal nodules or subependymal giant cell astrocytomas) and rigorous investigation using multiple experimental and bioinformatic approaches."

Referee 2: "In this study, Martin et al. collect an impressive amount of information regarding the genomic changes associated with tuberous sclerosis and transcriptional information that provides some insight into how these genomic changes lead to abnormal cell growth...The studies are comprehensive and the analyses appear to be well performed."

We thank the reviewers for their positive comments and have worked hard to make this an important publication in the field of tuberous sclerosis complex. We feel this manuscript will allow scientists to use the data presented here to facilitate basic, mechanistic, and translational research for this rare disease. We would be thrilled if these reviewers find our manuscript acceptable for publication in the *Nature Communications*.

Referee 1, comment 1: The justification for combining SEN/SEGAs was not clearly explained. The numbers for each of these tumor types should be more obvious in the methods. SEGA alone is used to describe some of the results and it is not clear if results from SENs were excluded to arrive at these conclusions.

Our study included 18 SEGA and 2 SEN, which we combined together into a single category referred to as "SEN/SEGA" throughout the text. Our primary reason for grouping these together is that they reflect a continuum of a single tissue type distinguished only by evidence of growth (SEGAs) or a lack of growth (SENs). In fact, these two lesions are indistinguishable histologically¹. We have now indicated this in the text (see *Results*, **page 5**). Also, because SEGAs are subject to surgical resection and SENs are not, we had far fewer SEN samples and felt they would be better represented with SEGAs than as a standalone category. In support of this, we were able to find a large conserved network of differentially expressed genes shared by SENs and SEGAs (Figure 4a).

Referee 1, comment 2, part I: It may be overreaching to conclude that all the underexpressed genes represented in Table 1 are important in normal kidney development and function. Also, please comment on the extent to which these results may reflect the admixture of different cell types in angiomyolipomas compared to normal kidney.

Indeed, the dramatic changes in gene expression (i.e., the magnitude of changes as well as level of conservation across RAs) are very likely caused by the unique admixture of cell types in RAs compared to the normal cellular composition of non-tumor kidney tissue. We have discussed that this cell type discrepancy may contribute to majority of the gene expression changes, especially in RA, in the *Discussion* section (see **page 14**). We also agree that it may be overreaching to conclude that all the biological processes in Table 1 are important for normal kidney development and function so to address this, we have modified our language in this section (see *Results*, **page 9**).

Referee 1, comment 2, part II: In the table legend, ["---" indicates no significant enrichment] does not appear in this table. In tables 1 and 2, it does not appear that the numbers of biological processes listed match the purported top 10.

This text, which was inadvertently copied from the Table 2 legend, has been removed from the legend of Table 1. We apologize for confusion with the number of enriched processes reported. We identified the enriched GO biological processes in each direction (increased and decreased expression compared to normal tissues) for each tumor type and then presented the top 10 (according to lowest FDR-adjusted *p*-values) with the exception of any processes with less than 10 matched genes. This led to just under 10 processes shown in the main tables. We have updated Table 1 and 2 legends to communicate this more clearly (see **pages 31-32**).

Referee 1, comments 3-5: In Fig 1c, blue bars indicate exons but there also appear to be regularly spaced blue bars that are shorter than the longer exon bars. In Fig 1d the key for deletion in the upper left corner should be moved further out of line with from the data so it is not confused with the results for 06-MG1. Fig 1e the Y-axis label for LRR 1.0 is missing. Fig 1m and 1n is confusing as labeled on x-axis, since it includes non-TSC tissue for first mark and does not indicate TSC tumors for second two marks, and the 1 and 2 are not immediately evident that they refer to the number of truncating mutations in the tumors. It is difficult to see the lines among the blue circles in 1m and these lines are not defined in the legend. CPM should also be defined in Fig 1 legend. In legend of Fig 2, define the tumor types abbreviated in 2b. In legend of Fig 3, define the 1* in 3b.

We have now eliminated the regularly-spaced blue bars in Figure 1c, which were base pair marks, and left only exon markers. We have moved the key in Figure 1d to avoid confusion, added the missing "1.0" label to Figure 1e, and updated the annotations in Figures 1m-n to improve clarity. We also changed the filled circles to open circles in Figures 1m-n to allow better visualization of the lines, which represent mean and standard deviation (now described in the legend). We defined CPM as counts per million in the legend of Figure 1 and defined each abbreviated malignant tumor type in the legend of Figure 2. Finally, we defined the "1*" in the legend of Figure 3, which represents a single TSC mutation tumor but which lacks data from targeted TSC-sequencing. We appreciate this detailed feedback as these corrections and clarifications improve the quality of the figures.

Referee 1, comment 6, part I: In the methods, the approach to defining germline and point mutations is defined and mixed together somewhat with results. It would be interesting to know the median and range for somatic point mutation VAF, not just that they were < 40% VAF.

Our goal was to use specific data to support our criteria for predicting the germline and somatic origin of mutations in tumors lacking matched normal tissues. This has led to a large section of the *Methods* devoted to this topic, which contains some results-like information. In addition to thresholds, we have now specified the medians and ranges for variant allelic fractions (VAFs) throughout this section (see **pages 20**).

Referee 1, comment 6, part II: Also in the methods, what defined positive FISH results when scoring 200 interphase nuclei?

FISH probes are routinely validated by counting nuclear signals in normal samples to determine a threshold (e.g, 2-3% of nuclei) above which a call of true gain or loss can be reasonably made (that is, a threshold which separates false from true positives). Importantly, as seen in Table S6, our FISH results were robust and never approached this threshold (the lowest percentage of nuclei containing an abnormal signal count used to make a gain or loss call was 32.4%).

Referee 1, comment 7, part I: In the discussion, the assertion that this is the first transheterozygous TSC1/TSC2 inactivation should consider similar results obtained on one tumor in a previous study (Hum Mol Genet. 2014;23:2023-9).

We appreciate the reviewer highlighting this previously published result. Indeed, in this report by Tyburczy *et al.* (2014), a periungual fibroma from a mosaic TSC2 patient was found to contain a low-frequency somatic TSC1 mutation. The authors noted the unexpected nature of this result and suggested it was unlikely that a mono-allelic mutation in TSC1 could cooperate with a germline TSC2 mutation to drive MTOR activation. We feel that together, our reports with similar observations from two independent patients lend credence to the idea that trans-heterozygous TSC1/TSC2 mutations may contribute to tumorigenesis in TSC. We have now discussed this reference in the *Discussion* (see **page 13**).

Referee 1, comment 7, part II: It is not clear how MTOR pathway alterations were assessed to conclude that "Surprisingly, we did not identify MTOR signaling, or anything related to this pathway, in our tumor-specific pathway analyses...".

Our primary pathway enrichment analysis was completed using GeneAnalytics which reported significantly enriched GO biological processes from up to 300 DEGs (increased or decreased expression from normal tissues). The fact that no pathways relating to MTOR signaling were found to be significantly enriched in this analysis in any lesion type led us to the conclusion quoted above. That said, we have now carefully reviewed the available MTOR pathways in the GO biological processes database and feel they may be inadequate to address this question. Therefore, we repeated our pathway enrichment analysis using MetaCore as an alternative approach and indeed, found that "signal transduction_mTORC1 upstream signaling" and "signal transduction_mTORC1 downstream signaling" were significantly enriched in SEN/SEGA (but not RA or cortical tuber) when using a less stringent DEG threshold criteria (\log_2 fold-change $\geq \pm 1$ and FDR-adjusted $p \leq 0.05$). This result is now described in the *Results* (**page 11**) and the data are now included in a new **Supplementary Table 13**.

Referee 2, comment 1: There are some areas that are not immediately clear regarding the mutations identified in TSC1 and TSC2. For example, the initial WES identified a majority of the mutations but it is not clearly described in the text of the results how many of these mutations were found in normal tissues versus abnormal tissues.

We have refined the text and reorganized this section to improve clarity (see *Results*, **pages 5-6**). We also recognize the complexity of the TSC1 and TSC2 mutational data and have included very detailed information in Supplementary Table 2 in order to provide readers with specific information that may be of interest for further analysis.

Referee 2, comment 2, part I: Categorizing cortical tubers as a "tumor" is misleading, and the authors note in the discussion that these lesions may be more appropriately categorized as a developmental malformation.

We appreciate that cortical tubers represent an abnormal tissue growth that is better defined as a developmental tissue malformation rather than *bona fide* tumor. When necessary throughout the manuscript, we have now incorporated the term "lesion" and "hamartoma," especially when referring to a cortical tuber or when referencing the entire group of abnormal growths we studied, which includes both tumors and cortical tubers. We believe this reduces confusion and improves the accuracy of our statements.

Referee 2, comment 2, part II: It appears that not all of the cortical tubers had targeted deep sequencing of TSC1/TSC2, and it would be interesting to know how read depth affected the likelihood of identifying a second hit mutation in cortical tubers versus other tissues. It would be interesting to

compare the allele frequencies of second hit mutations in RA/SEGAs versus cortical tubers. Are the allele frequencies of second hit mutations in cortical tubers low enough to suggest that deep sequencing is missing a significant portion of mutations that are below the alternative allele frequency detection limit? Or are alternate allele frequencies similar between RA/SEGAs and cortical tubes, suggesting that they should have been identified and that many cortical tubers can arise without a second hit mutation?

The reviewer raises an important point worth addressing. There were only five cortical tubers with second (somatic) TSC hits. Unfortunately, three of these were either CN-LOH or large deletions and the remaining two were atypical and complex mutations involving an apparent loss of intronic DNA (for which we have now provided additional clarity and references – see *Discussion*, **page 13**), none of which allow accurate quantification of variant allelic fractions. Instead, we retrieved the variant allelic fractions (VAFs) for all somatic variants detected (that is, non-*TSC1/TSC2*) in cortical tubers, SEN/SEGA, and RA. Importantly, the mean VAF for somatic mutations in all tumor types was similar (18-20%) (see **Referee Figure A**). This suggests that somatic variants generally occur at similar frequencies in cortical tubers and other TSC lesion types and therefore, we would not expect somatic *TSC1/TSC2* mutation in tubers to be any less frequent than in the other lesions. It is also important to point out that our workflows were able to identify mutations as low as 2% VAF, regardless of tissue type. Moreover, our low frequency variant pipeline was able to identify variants as low as 0.5% frequency, and while we found these to be false positives, it suggests that if real variants existed in the tubers at these exceptionally low frequencies, they would have been detected.

Referee 2, comment 3: It would be interesting to analyze the breakdown of tissues where chromosomal number abnormalities were identified to know whether certain tissues are more or less likely to have rearrangements. For example, only one cortical tuber demonstrated a CNA, whereas several RA/SEGAs were identified, and it would be interesting to know whether cortical tubers are significantly less likely to have CNAs.

We identified CNAs in each of the three major lesion types studied as well as CRMs at varying frequencies (this is now made clear in the *Results* (see **page 8**). Indeed, a lower frequency of cortical tubers contained CNAs (1 of 31; 3%) compared to RAs (3 of 20; 15%), SEN/SEGAs (4 of 19; 21%), and CRMs (1 of 5; 20%). However, we failed to find statistically significant differences between the CNA frequency of cortical tubers and all other TSC lesions using two-tailed Fisher's exact tests ($p = 0.08$).

Referee 2, comment 4: The methylation studies identify many genes with hypermethylated promoters, and about 10% of these genes were found to be differentially expressed with RNAseq. It would be interesting to know whether this is a significant enrichment in differentially expressed genes or could have been arrived at by chance alone.

To address this question, we performed a chi-square test for independence, which failed to support a statistically significant association between hypermethylation status and identification of a gene as differentially expressed [$\chi^2(1, N = 16,408) = 1.873, p = 0.17$]. That said, the decreased expression of 12 of the 13 (92%) genes both hypermethylated and differentially expressed in RAs supports that hypermethylation contributes to biologically relevant transcriptional silencing. We have added the result of this test and discussion to *Results* (**page 9**).

Referee 2, comment 5: The differential expression analysis states that the criteria were $\log_2 > +/- 2$. Is this really the case (i.e. requiring a fold change >4) or is it really $\log_2 > +/- 1$? In addition, the authors state that the other criteria for differential expression is a p-value less than 0.001. It is unclear whether this p-value is corrected for multiple comparisons, and

if it is not, it would be important to know what the estimated false discovery rate at that p-value is.

Indeed, we used a log₂ fold-change $\pm \geq 2$ (fold-change $\pm \geq 4$) as a threshold to identify differentially expressed genes (those listed in Tables S8-10 and which fueled the pathway enrichment analyses reported in Tables 1-2). The abundance of genes identified that exceed this threshold underscores the magnitude of expression changes observed in each tumor type when compared to normal tissue counterparts. In addition, we did correct for multiple comparisons in this analysis by applying a false discovery rate (FDR) adjusted p-value threshold of ≤ 0.001 . We have reviewed the text and clarified the adjusted nature of p-values used throughout.

Referee 2, comment 6: The authors use CIBERSORT to evaluate the likely cellular composition of the tissue based on the gene expression patterns that are present. As the authors point out, this method is only as good as the data for the cell types of interest. The authors identify many differentially expressed genes that are associated with inflammation, and they suggest that this is due to inflammatory infiltrates. This is shown using IHC in a SEGA, but is it possible that reactive astrocytosis could lead to some of these changes in cortical tubers without lymphocyte infiltration?

Our CIBERSORT analysis detected a fraction of leukocytes in SEN/SEGA (and to a lesser degree, cortical tubers) that was absent in normal brain tissue (Figure 5a). Specifically, this was predicted to largely represent activated macrophages (Figure 5c). As the primary brain-resident macrophage, we expect this signature to be driven largely by microglia, a prediction supported by IBA1/AIF1 staining. We now recognize that we used the terminology “leukocyte infiltration,” which inappropriately suggested we were referring to immune molecules infiltrating the brain from circulation (i.e., through the blood-brain barrier). While evidence for this type of adaptive immune response has been recently documented in certain cases of neuroinflammation, the brain is considered a largely immune privileged organ, and as such, it is affected primarily by innate immune processes involving activation of brain-resident macrophages (microglia) and astrocytes (into “reactive astrocytes”), among other cell types. We do agree it is likely that reactive astrocytosis is occurring, in addition to microglia activation, which we have now indicated in the text (see *Discussion*, **page 15**). We feel that refinement of our language in the text when discussing the inflammation observed in association with SEN/SEGA and cortical tubers has improved the clarity of the data in this section (see *Results*, **page 11**) and should eliminate confusion as to the type of immune response we are referencing.

Referee 2, comment 7: It might be interesting to note if and how many of these patients received mTOR inhibitors and whether this had any appreciable effect on the gene expression patterns.

We used 42 lesions for gene expression studies which could have been exposed to MTOR inhibitor therapies. Of these, 27 (64%) were collected from a biobank and associated with deidentified records with limited clinical annotation, so we were unable to determine whether patients were exposed to MTOR inhibitors. It should be noted that nearly all (25/27) of these biobank samples were harvested prior to 2010, when clinical use of mTOR inhibitors was first approved in the United States for TSC. Of the remaining non-biobank cases, 13 (31%) were from patients confirmed to have not received MTOR inhibitors prior to tissue collection. The remaining 2 samples (5%) were lesions from patients who did receive everolimus or sirolimus, however, therapy was withheld for 6 to 21 days prior to surgical resection. Taken together, we feel an analysis considering MTOR inhibitor status a variable insufficiently powered. From the information we were able to gather, we are assured that the majority of our lesion samples were MTOR inhibitor naïve and thus, pharmacological inhibition of MTOR should not play a dominant role when interpreting the gene expression results.

Referee 2, comment 8: In the introduction, the authors state that several of the manifestations of TSC such as epilepsy and autism are due to hamartomas or cortical tubers. However, there are a number of patients with TSC who do

not have tubers but do have epilepsy. Either they have lesions that are below the resolution of current imaging techniques or haploinsufficiency of TSC1/2 is sufficient to lead to epilepsy. This would be worth adding to the discussion.

We appreciate this feedback and have now added mention of this type of haploinsufficiency to our *Discussion* (see **page 13**).

Referee 2, comment 9: In figure 1m, the reduction in TSC2 mRNA appears very small. How do the authors interpret this result and its clinical significance?

It should be emphasized that the Y-axes in Figures 1m and 1n are log₂ transformed counts per million (CPM) expression values. In fact, lesions with two truncating TSC2 mutations had (on average) an approximate ~50% reduction in TSC2 transcripts from non-TSC control tissues, while tumors with two truncating TSC1 mutations had on average 70% lower TSC1 expression than non-TSC tissues.

Referee 2, comment 10: Based on Figure 2b, the authors state that "This mutation rate is substantially lower than almost all malignant tumor types, with the exception of acute myeloid leukemia (AML)." Since somatic mutation frequency increases exponentially with patient age, how do the authors' results change once you factor in the age of the patient? In other words, is the mutation rate in TSC lesions lower than that of non-TSC tumors in patients of similar age?

First, it should be noted that the median age of patients comprising the AML category in Figure 2b is 57 years (range: 18-88 years)². Second, we appreciate that an association between patient age and mutational burden has been established for malignant tumors. The median age of the TSC patients (at the time of sample collection) we used to calculate somatic mutational burden was 8 years (range: 0-38 years) and the somatic mutation rate was 0.31 mutations per Mb. Milholland and colleagues (2015) stratified TCGA patients by age and reported that malignant tumors from patients under 20 years of age (their youngest category) had 0.37 mutations per Mb (95% CI = 0.30 to 0.43)³. Qualitatively, this suggests our low mutational burden is consistent with that of similarly young patients in the TCGA. To further address the relationship between age and mutational burden within our cohort, we plotted TSC patient age (at time of sample collection) by mutational burden in an XY plot and did not observe a positive correlation (see **Referee Figure B**). In fact, 5 of the 6 most highly mutated lesions in our cohort (≥ 1 mutation per Mb) came from patients less than 10 years of age. Finally, we searched for the contribution of known mutational signatures in these six highly mutated TSC tumors and found no evidence of involvement of two age-related signatures (signatures 1A and 1B) (see **Referee Figure C**)⁴. We expanded this to all TSC tumors regardless of mutational burden and while samples from the 3 oldest patients in our study did show some influence of age-related signatures, there was no enrichment in these signatures with patient age overall (see **Referee Figure D**).

Referee Figure. (A) The somatic variant allelic fractions (VAFs) of major TSC lesion types are shown (symbol: mean; error bars: standard deviation). (B) The somatic mutation rates of individual lesions are plotted on the Y-axis with patient age (at time of sample acquisition) plotted on the X-axis. (C) The relative contributions of individual mutation signatures (1-21) from the 6 samples with highest mutation burdens are shown from low (blue) to high (red) according to the color legend. The age-related signatures (1A and 1B) are noted in red. (D) The relative contribution of mutational signatures 1A and 1B is shown for all TSC lesions. The age of patients at time of lesion acquisition is shown on the bottom (youngest (white) to oldest (green)).

References:

1. Goh, S., Butler, W. & Thiele, E.A. Subependymal giant cell tumors in tuberous sclerosis complex. *Neurology* **63**, 1457-61 (2004).
2. Cancer Genome Atlas Research, N. Genomic and epigenomic landscapes of adult de novo acute myeloid leukemia. *N Engl J Med* **368**, 2059-74 (2013).
3. Milholland, B., Auton, A., Suh, Y. & Vijg, J. Age-related somatic mutations in the cancer genome. *Oncotarget* **6**, 24627-35 (2015).
4. Alexandrov, L.B. *et al.* Signatures of mutational processes in human cancer. *Nature* **500**, 415-21 (2013).

REVIEWERS' COMMENTS:

Reviewer #1 (Remarks to the Author):

Concerns have been addressed. I am pleased with the ways the authors have provided additional data in support of their conclusions. Please check some of the supplementary table legends for proper capitalization (e.g. LoFreq, Human Genome Variation Society). This is the most comprehensive analysis of TSC tumors to date. It is a benchmark paper for the field.

Reviewer #2 (Remarks to the Author):

I am satisfied with the changes that the authors have made in response to the reviews.

Manuscript ID # NCOMMS-16-28761B

Response to Reviewers

We appreciate the time spent on this manuscript by our reviewers, and are extremely pleased that they feel our response to the revision requests have been adequately addressed.

Reviewer #1 (Remarks to the Author):

Concerns have been addressed. I am pleased with the ways the authors have provided additional data in support of their conclusions. Please check some of the supplementary table legends for proper capitalization (e.g. LoFreq, Human Genome Variation Society). This is the most comprehensive analysis of TSC tumors to date. It is a benchmark paper for the field.

We appreciate this feedback. We have also carefully edited the Supplementary Table (and Supplementary Data) legends for proper capitalization.

Reviewer #2 (Remarks to the Author):

I am satisfied with the changes that the authors have made in response to the reviews.

We are pleased that Reviewer #2 is satisfied with our response to the initial reviews.